# The Intracellular Proteome as a Source for Novel Targets in CAR-T and T-Cell Engagers-Based Immunotherapy

**DOI:** 10.3390/cells12010027

**Published:** 2022-12-21

**Authors:** Inbar Arman, Maya Haus-Cohen, Yoram Reiter

**Affiliations:** Laboratory of Molecular Immunology and Immunotherapy, Faculty of Biology, Technion–Israel Institute of Haifa, Haifa 320003, Israel

**Keywords:** T-cell-receptor-like antibodies, T cell receptor mimic, chimeric antigen receptor, peptide, MHC

## Abstract

The impressive clinical success of cancer immunotherapy has motivated the continued search for new targets that may serve to guide potent effector functions in an attempt to efficiently kill malignant cells. The intracellular proteome is an interesting source for such new targets, such as neo-antigens and others, with growing interest in their application for cell-based immunotherapies. These intracellular-derived targets are peptides presented by MHC class I molecules on the cell surface of malignant cells. These disease-specific class I HLA–peptide complexes can be targeted by specific TCRs or by antibodies that mimic TCR-specificity, termed TCR-like (TCRL) antibodies. Adoptive cell transfer of TCR engineered T cells and T-cell-receptor-like based CAR-T cells, targeted against a peptide-MHC of interest, are currently tested as cancer therapeutic agents in pre-clinical and clinical trials, along with soluble TCR- and TCRL-based agents, such as immunotoxins and bi-specific T cell engagers. Targeting the intracellular proteome using TCRL- and TCR-based molecules shows promising results in cancer immunotherapy, as exemplified by the success of the anti-gp100/HLA-A2 TCR-based T cell engager, recently approved by the FDA for the treatment of unresectable or metastatic uveal melanoma. This review is focused on the selection and isolation processes of TCR- and TCRL-based targeting moieties, with a spotlight on pre-clinical and clinical studies, examining peptide-MHC targeting agents in cancer immunotherapy.

## 1. The Source of Intracellular Derived Targets

Adaptive immune surveillance mechanisms relay on cell surface presentation of MHC class I and II molecules, bound to peptides derived from either intracellular or extracellular proteins, respectively. Presentation of non-self-intracellular peptides, on MHC class I molecules to CD8+ cytotoxic T cells via a T cell receptor (TCR), allows the recognition of viral infected or malignant cells, resulting in target-specific immune response, aiming for abnormal cell elimination [1].

All nucleated cells express MHC class I molecules, a heterodimer containing a non-polymorphic β2 macroglobulin (β2M) light chain and a polymorphic heavy chain, encoded in humans by three duplicated genes on chromosome 6, named HLA-A, HLA-B, and HLA-C [2]. The MHC class I heavy chain was found to be the most polymorphic gene in humans, where each individual expresses a unique set of six HLA heavy chains out of over 20,000 polymorphic alleles [3]. 

Assembled MHC class I heterodimers mainly differ from one another in their peptide binding groove domains, resulting in a variable set of peptides that, based on their charge and length, may fit a certain MHC class I variant, but not others [4]. Accordingly, certain diseases were found to be positively correlated with specific MHC class I alleles, such as HLA-B*27:05 in ankylosing spondylitis [5]. Moreover, a change in a single amino acid within the peptide binding groove may completely alter the HLA presented peptide repertoire. For example, a dramatic change in peptide binding repertoire was observed between HLA-B*15:01 and HLA-B*15:18 alleles, which only differ in a single amino acid [6]. Of note, some HLA class I heavy chains were found to be more abundant than others, such as the presentation of HLA-A2 allele in approximately 50% of Caucasian humans [7]. 

Proteins that originated in either the cytosol or nucleus are degraded by the 26S proteasome, followed by presentation on MHC class I molecules [8]. The source of 26S degraded peptides includes (a) defective ribosomal products (DRIPs), consisting mainly of misfolded proteins and prematurely terminated polypeptides degraded in close to synthesis time; (b) proteins at the end of their lifetime; and (c) signal sequence fragments. 26S Degraded peptides translocate to the ER lumen by transporter associated with antigen processing (TAP), where they interact with the MHC class I molecule [9]. 

Assembly of peptide/MHC class I complex is a multistep process, starting with the temporally stabilization of the MHC class I heterodimer, containing the β2M light chain and the variable heavy chain, by ER chaperones such as calreticulin, ERp57, PDI, and tapasin, creating the peptide loading complex (PLC) [10]. Then, the usually 8–12 amino acid (aa) long peptide bound to TAP interacts with tapasin, delivering the peptide to the ER lumen. Release of all MHC chaperones occurs if the delivered peptide binds the MHC class I peptide binding groove with sufficient affinity, creating a stable peptide/MHC class I complex [11]. During assembly, a certain MHC class I molecule may switch several different peptides, resulting in a conformational change in each peptide exchange. ER release and transport to the cell surface occur when a high affinity peptide binds the MHC binding groove, creating a stable peptide/MHC class I complex [12]. Peptides that failed to stabilize the MHC class I molecule are transformed back to the cytosol for full degradation via the ER-associated degradation (ERAD) pathway [13]. An illustration of the pMHC assembly process can be seen in Figure 1. 

Presentation of peptide/MHC class I molecules on the cell surface was found to be influenced by several factors, such as protein copy number, intracellular conditions, HLA heavy chain allele variants, and cell type [1,13]. Proteins expressed at a copy number lower than 1000 may not be presented on MHC class I molecules owing to their relatively low expression. Consequently, these proteins, especially if unstable, may evade MHC-based immune surveillance system [14]. Moreover, different intracellular conditions may affect protein expression, such as upregulation of protein processing and MHC class I expression in the presence of IFNγ and enhanced protein degradation in irradiated treated cells, affecting peptide repertoire presented on the cell surface [15]. Of note, the cell surface presentation level of MHC class I molecules varies between 10,000 and 500,000 molecules, depending on cell type. For example, professional antigen presenting cells, such as DCs, express high MHC class I levels, while some malignant cells downregulate MHC class I presentation, escaping immune surveillance [16,17].

## 2. Targeting Intracellular Proteins

Interaction between T cell receptor (TCR) and cell surface foreign or abnormal peptides, in the context of MHC class I molecules, provides one of the first and most critical steps in adaptive immune response [18]. Mature αβ T cells express a disulfide linked heterodimeric TCRs, consisting of the TCRα and TCRβ chains [19]. Unique TCR sequences are assembled owing to somatic recombination in variable (V), diversity (D), and joining (J) regions, creating the CDR1, CDR2, and CDR3 domains of the TCR antigen binding site [20]. Recognition and elimination of specific malignant and infected cells by the adaptive immune arm, through the unique TCR-pMHC interaction, is dependent on a variety of molecular and cellular features such as variation in MHC alleles, intracellular processing by the proteasome, peptide presentation, and TCR clonality [21]. TCR and pMHC complex interactions induce intracellular T cell signaling via immune-receptor tyrosine-based activation motif (ITAMs), located on the CD3 chains expressed as part of the TCR complex. Affinity between TCR and pMHC complex affects the intracellular signaling level, thereby influencing T cell faith. For example, it was found that a strong interaction of CD8+ TCRs with pMHC molecules usually results in robust T cell proliferation, while weak interaction results in CD8+ memory T cell formation [22]. 

Targeting an intracellular abnormal protein-derived peptide, expressed in the context of MHC class I molecule, can be achieved using engineered T cells, manipulated to express specific TCRα and TCRβ genes, with specificity toward a desired pMHC complex. In this case, target cells must express the peptide of interest in the context of a specific MHC class I variant, as engineered TCR recognizes both peptide and MHC class I molecules. Accordingly, TCR-based treatments focus on relatively abundant MHC class I alleles, such as HLA-A*02:01 [23]. Tumor targeting TCR sequences can be identified via isolation and deep sequencing strategies of tumor infiltrating lymphocytes (TILs). These TIL CD8+ lymphocytes, present in the tumor environment, are sorted as single cells, followed by sequencing and TCRα-TCRβ pairing analysis. Alternatively, tetrameric cancer-related peptide-MHC molecules can be used to identify peripheral blood lymphocytes (PBLs) expressing TCRs in diseased patients. These TCRs can further be cloned and re-expressed in cytotoxic T lymphocytes (CTLs) derived from peripheral blood, creating tumor-specific T cells [24,25]. For example, TCRs targeting melanoma-associated antigen recognized by T cells (MART-1) in the context of HLA-A2 derived from melanoma metastatic patient TILs were cloned and transduced into activated T cells, showing anti-tumor activity against HLA-A2+ melanoma cells [26]. Optimization of TCR affinity toward a desired antigen also improved the therapeutic potential of TCR-based treatment, such as the affinity optimized AFP/HLA-A2 targeting TCR, showing improved activity against liver malignant cells [27]. 

Mimicking adaptive immune response surveillance system to target abnormal peptide presentation can also be achieved using T-cell-receptor-like (TCRL) antibodies, also termed T cell receptor mimic (TCRm) antibodies. In this approach, combining the advantages of the both humoral and cellular adaptive immune response, antibodies mimicking the binding of a TCR to pMHC class I complex are used to target abnormal peptides presented on the cell surface of malignant or infected cells with nanomolar affinities [28,29]. The antigen binding domain of these antibodies can further be cloned to create fusion molecules, such as immunocytokines, bispecific T-cell engager (BiTE), and chimeric antigen receptor (CAR)-T constructs [30]. The first step in TCRL isolation usually requires the expression and purification of the target pMHC of interest. This can be achieved by either expression of single chain trimers (SCTs), which is a single polypeptide consisting of all three subunits of the pMHC class I molecule including β2M, HLA heavy chain, and the peptide of interest, all connected with flexible linkers, or by β2M and HLA heavy chain linked covalently with a flexible linker, expressed in E. coli, and subsequently refolded with the desired MHC-restricted peptide. Alternatively, MHC class I–peptide complexes can also be generated using separate β2M and HLA heavy chains, expressed in E. coli, followed by refolding with a synthetic peptide, thus obtaining the full tetrameric pMHC class I–peptide complex. Once correctly folded, these constructs resemble the cognate pMHC class I molecules presented on cells, in a soluble form [31,32]. Traditional methods to isolate TCRL antibodies include scanning of phage display libraries or in vivo immunizations with the pMHC complex, followed by hybridoma or human B cell cloning assays. Using the phage display method, ScFV- or Fab-based libraries are scanned against monomeric soluble target pMHC molecules and control pMHC complexes. ScFV/Fab that showed specificity toward target pMHC can be further cloned into Full IgG abs [33]. For example, the isolation of a TCRL ab against the tumor antigen WT1 in the context of HLA-A2 was achieved using ScFV-based phage display library scan [34]. In TCRL isolation-based hybridoma assay, animals are immunized with specific pMHC molecule or antigen presenting cells (APCs), expressing the pMHC of interest [35]. Vaccine triggered immune response may result in antibodies that are specific toward the pMHC molecule. Using vaccines based on pMHC expressing cells was found to be superior to recombinant soluble pMHC molecule vaccination, such as the isolation of TCRL against PRI/HLA-A2 antigen, tested in both soluble and cell vaccine-based approaches using Balb/c mice. Immunization with pMHC soluble molecules resulted in several pMHC targeting potential clones, while using vaccination-based PRI pulsed cells resulted in no potential clones [36]. B cell cloning is another vaccine-based approach for the purpose of pMHC TCRL isolation. In this case, animals are immunized with the target of interest, followed by single-cell sorting of PBMCs, based on fluorophore labeled target tetramers and B cell markers. Ig primers are then used to amplify cDNAs derived from sorted cells, followed by cloning into full IgG sequences [37]. Recently, Tatsuhiko et al. described a new chip-based TCRL isolation method. Here, pMHC rabbit immunization followed by single-cell chip-based selection of pMHC specific antibody producing cells derived from target organs was subjected to amplification of heavy and light cDNA, resulting in the isolation of specific TCRL abs [38,39]. A summary of TCRL and TCR isolation methods can be seen in Figure 2.

## 3. TCR and TCR-Like Constructs as Therapeutic Agents

### 3.1. Choosing the pMHC of Interest

The choice of p/MHC target for a TCRL- or TCR-based therapy is a key factor in the design and application of the therapeutic agent as it will dictate the delicate balance between an unmet need indication, efficacy, and the key issue of selectivity and specificity. The targeted pMHC complex should preferably be a neo-antigen, expressed exclusively on malignant cells. These tumor specific antigens (TSAs) result from specific mutations in the malignant cells, thus targeting these unique and specific pMHC targets increases treatment safety by increasing on-target and reducing off-tumor responses. For example, specific in vitro activity was detected in TCRs isolated from CML patients, directed against the crucial BCR-ABL mutation, resulting from a translocation between chromosome 22 and 9, found in approximately 95% of CML patients [40]. Alternatively, tumor-associated antigens (TAAs), such as growth factors and cell cycle oncogenic proteins, which are known to be largely expressed on malignant cells, but their expression may be found in other tissues, can be targeted as well, while considering possible on-target but off-tissue response [41]. Prediction of peptides formed by degradation of a protein of interest and their binding to MHC class I molecule can be predicted by dedicated algorithms [42]. Alternatively, such TAAs and TSAs targets can be identified and characterized biochemically by elution of peptide presented on malignant cells in the context of MHC class I molecules, followed by mass spectrometry [35]. The quest for new targets continues, with a current phase 1 clinical trial for the identification of somatic mutations and HLA typing in several solid tumors. Targeting a pMHC derived from a protein that has a key role in cell proliferation, such as growth factors, would potentially improve the chances that the target pMHC will not be lost or down modulated under selection pressure [43]. Moreover, choosing a relatively abundant MHC class I allele is also a key factor in target selection, as treatment is relevant only in the context of the specific peptide on the specific MHC class I allele. Accordingly, as HLA-A2:01 was found to be the most abundant out of the HLA-A alleles in Caucasians, it is usually selected as the MHC allele in targeted TCRL- and TCR-based therapy [44]. An improvement in TCR- or TCRL-CAR-based therapy can also be achieved by targeting high-density pMHC tumor-expressing cells, as the relative presentation of pMHC of interest on malignant cells was found to be positively correlated with T cell response [45]. Illustration of different TCRL- and TCR-based molecules can be found in Figure 3. 

### 3.2. TCRL-Based Soluble Molecules

Targeting malignant cell intracellular proteome using extracellular recognition of pMHC complex by soluble molecules can be achieved using TCRL naked antibodies, soluble TCRs, armed TCRL-/TCR-based immunocytokines and bi-specific proteins; each tool provides a unique approach for tumor cell recognition and elimination [35,46].

### 3.3. TCRL- and TCR-Based Soluble Non-Armed Molecules

Soluble naked full IgG TCRL antibody can target and induce specific tumor cell elimination via several mechanisms, such as (a) antibody dependent cellular cytotoxicity (ADCC), where innate immune cells expressing Fc-gamma receptor binds the constant IgG region of the antibody, inducing specific lysis of the ab bounded target cell [47]; (b) complement dependent cytolysis (CDC), where antibody bounded target cells undergo lysis via C5b-9 membrane attack complex [48]; and (c) antibody-dependent phagocytosis (ADCP), mainly via target cell internalization and lysis by macrophages, mediated by Fc gamma receptor [49]. Activation of both ADCC and CDC mechanisms was observed using a TCRL antibody, named 8F4, targeting PRI/HLA-A2 expressed on the cell surface of acute myeloid leukemia (AML) progenitor cells [36]. Upon stability enhancement, soluble TCRs can also be used to target malignant cells. Several approaches were used to adjust the TCR native form to a stable soluble agent, such as random mutagenesis and computational modeling, aiming for surface hydrophobic residues’ replacement. Alternatively, fusion of the Ig constant domain to the TCRα and TCRβ extracellular domain, jun-fos leucine zipper fusion to the TCR C terminus, or novel addition of the interchain disulfide-bridge were found to improve soluble TCR stability [50]. Several different soluble TCR molecules, mutated with an intradomain disulfide bond, showed superior stability in comparison with non-mutated soluble TCRs and maintained binding specificity toward pMHC-expressing target cells [51].

### 3.4. Bi-Specific TCRL- and TCR-Based Soluble Molecules

Malignant and immune cells dual targeting, can be achieved using bi-specific agents. In this approach, TCRL- or TCR-based molecules, targeted toward pMHC presented on malignant cells, can be fused to an anti-CD3 effector moiety, redirecting T cells against target malignant cells [52]. Recently, a TCR-based bi-specific molecule, termed KIMMTRAK by Immunocore, targeted toward the malignant cell surface marker gp100/HLA-A2, fused to an anti-CD3 effector domain, was approved by the FDA for the treatment of unresectable or metastatic uveal melanoma, showing clinical benefit in a phase III clinical trial, via recruitment of T cells to gp100/HLA-A2 positive cells [53,54]. Of note, this drug is currently under phase 1/2 clinical trial for the treatment of Cutaneous melanoma [55]. Several pMHC binding bi-specific molecules are currently under clinical trials, such as IMA401 and IMA202 by Immatics, targeting MAGE-A4/8 and MAGE-A1 in the context of HLA-A2, respectively [56], and HLA-A2-WT1 X CD3 bi-specific molecule for the treatment of adult acute myeloid leukemia by Roche/Genentech [57].

### 3.5. Armed TCRL- and TCR-Based Soluble Molecules

Selective delivery of cytotoxic agents to tumor cells can be achieved using antibody–drug conjugates (ADCs), which can induce specific apoptosis of target cells, reducing the risk and severe side effects of general administration of cytotoxic materials [58]. Such a TCRL armed molecule was tested against MART-1/HLA-A201 melanoma positive cells, delivering the PE38KDEL toxin to malignant target cells. This immunotoxin was found to significantly and specifically inhibit human melanoma growth in severe combined immunodeficient mice [59]. Another immunotoxin, targeting the upregulated P53 peptide presented in the context of HLA-A24 on malignant cells, conjugated to a toxic DNA alkylating agent was found to limit tumor growth in NSG xenograft model [60].

### 3.6. TCR-Engineered T Cells

Engineered T cells, expressing a specific TCR, can be used for recognition and elimination of malignant cells, eliciting cytotoxic activity in response to pMHC-expressing target cells [61]. The functional avidity of engineered T cells is a crucial measurement in the selection process of a TCR, assessed by the in vitro response of TCR-engineered T cell to different concentrations of the pMHC target epitope. In vitro assays determining functional avidity include the EC50 peptide concentration required for cytokine secretion, cytotoxic activity, and T cell proliferation [62]. Functional avidity was previously shown to be improved using either TCR framework mutations or co-transfection of the TCR with the four CD3 chains [63]. TCR affinity, measured by the interaction strength between pMHC and a single TCR, is also an important parameter in TCR candidate selection, as very high affinity TCRs may have impaired function owing to intense but short-lived response, while low TCR affinity may result in superior anti-tumor activity [64,65]. The relative expression of target epitope on malignant cells, termed antigen density, can also influence TCR activity, as relatively high antigen density results in impaired T cell activity [66]. Additional challenge in TCR engineering is the concern for inappropriate α/β TCR chains pairing, as transduced T cells also express their native TCR, which may lower the desired TCR expression and, as a consequence, lower treatment efficiency. Several groups developed tools to address this problem, For example, Cohen et al. showed that desired TCR pairing can be achieved using engineered TCRs with constant murine regions, which preferably bind each other rather than the endogenous human TCR chains. Alternatively, the addition of a second disulfide bond to the transduced TCR can also improve the desired TCR pairing [67,68]. Another approach is to construct a single-chain TCR, which consists of both of the variable α and β TCR regions, connected by a linker [69]. Extensive work in the TCR-engineered T cell field resulted in the first documented TCR-based clinical trial, targeted against melanoma antigen recognized by T cells 1 (MART-1), showing promising results, including tumor regression. These encouraging results led to the development of several anti-cancer TCR-based treatments, against targets such as gp100, NY-ESO-1, and AFP [70]. Currently active clinical trials, found in ClinicalTrials.gov, targeting cancer epitopes via TCR-engineered T cells are summarized in Table 1. 

### 3.7. TCRL-Based CAR Engineered T Cells

CAR-T cells are synthetic antigen-specific constructs expressed on lymphocytes, programmed to recognize and induce cancer cell death. Classical CAR construct consists of an extracellular domain, which is usually based on an ScFV structure. This region is composed of a short flexible linker, connecting the VH and VL domains of a monoclonal antibody, dictating CAR specificity toward a desired antigen. Traditional CAR constructs target MHC independent epitopes on the cell surface of malignant cells, such as the 2017 FDA approved anti-CD19 CAR-T cell therapy, showing remarkable success against B cell malignancies [71,72]. Expending CAR-T targets to intracellular epitopes can be achieved using TCRL-based CAR-T cells, directed against MHC dependent neoepitopes, based on the VH and VL regions of a TCRL antibody [73]. As in TCR-engineered T cells, CAR-T activity depends on several factors, such as CAR-T avidity, affinity, and the antigen density, expressed by target malignant cells, all influencing CAR-T efficiency in tumor elimination. Comprehensive research, examining the influence of affinity, avidity, and antigen density using TCRL-based CAR-T cells, targeting Tyr/HLA-A2 antigen, was recently published by our group. In this study, we found that, as in TCR activity, very high antigen density results in diminished CAR-T activity, measured by cytokine production and CD107a upregulation. We also found that an optimal CAR-T response correlates with native TCR activity, at an antigen range of approximately 10^7^–10^5^ M, showing similar results when evaluating CAR-T with different avidities and affinities [74]. Interestingly, cumulative results suggest that maximal CAR-T activity does not correlate with high affinity, as CAR-T cells with intermediate affinities (16–35 nM) showed an improved T cell response [74,75]. Currently, there are no public documented clinical trials examining the efficacy of TCRL-based CAR-T cells, but extensive work can be found in pre-clinical studies, such as the CAR-T targeting PR1/HLA-A2 epitope, derived from leukemia-associated antigen proteinase 3 and neutrophil elastase, showing specific activity against primary AML blasts, in an HLA-A2-dependent manner [76]. Another example is the anti-melanoma CAR-T construct targeting NY-ESO-1 peptide in the context of HLA-A2. This CAR was tested for the treatment of melanoma implanted mouse model, resulting in delayed tumor progression. Of note, CAR targeting domain was originally based on an anti-NY-ESO-1/HLA-A2 Fab, which apparently, owing to high avidity, was found to lose specificity toward PR1/HLA-A2 epitope as a CAR, showing activity against non-peptide dependent HLA-A2 positive cells. Based on a crystal structure, specific mutation implemented to lower construct affinity toward the HLA-A2 epitope successfully resulted in a specific anti NY-ESO-1/hla-a2 CAR construct [77]. Additional TCRL-based CAR-T examples targeting hematological malignancies and solid tumors can be found against the intracellular onco-protein WT1, specific liver cancer marker alpha-fetoprotein (AFP), and gp100, all showing promising results in pre-clinical studies [78,79,80].

## 4. Concluding Remarks

Although pMHC targeting moieties show promising results in tumor eliminations, there is still limited information regarding the potential of these therapeutic agents in preclinical and clinical studies. Accordingly, TCRL- and TCR-based soluble molecules, engineered TCRs, and TCRL-based CAR-T cells should be further examined for their safety and potential unpredictable cross reactivity toward non-malignant pMHC-expressing cells [81]. Another challenge in targeting pMHC epitopes is the identification of common neo-peptides, shared among a large group of patients. This limitation may be addressed by experimental analysis and further characterization of prevalent pMHC complexes [82]. Finally, immune escape mechanisms, such as the downregulation of pMHC expression found in malignant cells, should be taken under consideration when treating patients with pMHC-targeting molecules [83]. Despite these mentioned limitations, current publications demonstrate compelling evidence and future promise for TCRL- and TCR-based therapies as potential new therapeutic modalities in cancer immunotherapy. 

## Figures and Tables

**Figure 1 cells-12-00027-f001:**
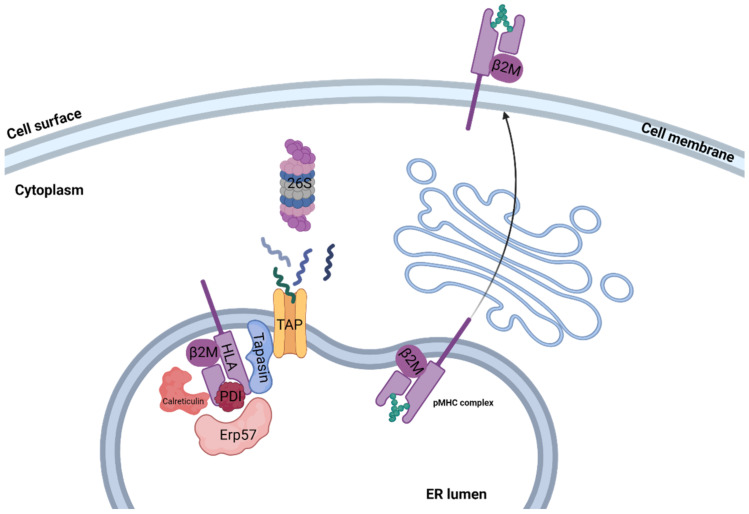
Class I pMHC presentation. Illustration of antigen processing and MHC class I presentation classic pathway. Proteins that originated in either the cytosol or nucleus are degraded to short peptides by the 26S proteasome. Peptides are then transported to the ER lumen by TAP, where they interact with the MHC class I molecule. Release of the PLC, which temporarily stabilizes the MHC class I complex, occurs if the delivered peptide binds the MHC class I peptide binding groove with sufficient affinity, creating a stable peptide/MHC class I complex. The stable pMHC complex is then transported to the cell surface, where it may interact with CD8+ T cells.

**Figure 2 cells-12-00027-f002:**
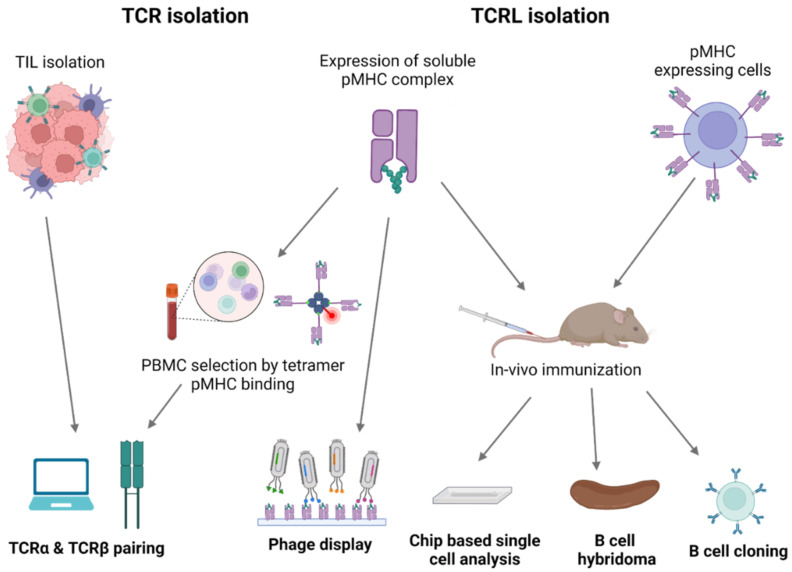
TCR and TCRL isolation methods. Illustration of TCR and TCRL identification and isolation process. TCR sequences targeting tumor-related pMHC dependent epitopes can be identified via isolation of TILs, present in the tumor environment, or via flow cytometry analysis of PBMCs, using fluorophore labeled pMHC tetrameric complexes. In both cases, computational analysis is required for TCRα and TCRβ pairing. TCRL isolation can be achieved using phage display libraries, where phages are scanned against the pMHC of interest or by in vivo immunization, using recombinant pMHC molecule or pMHC expressing cells, followed by hybridoma analysis, B cell cloning, or chip-based single cell analysis.

**Figure 3 cells-12-00027-f003:**
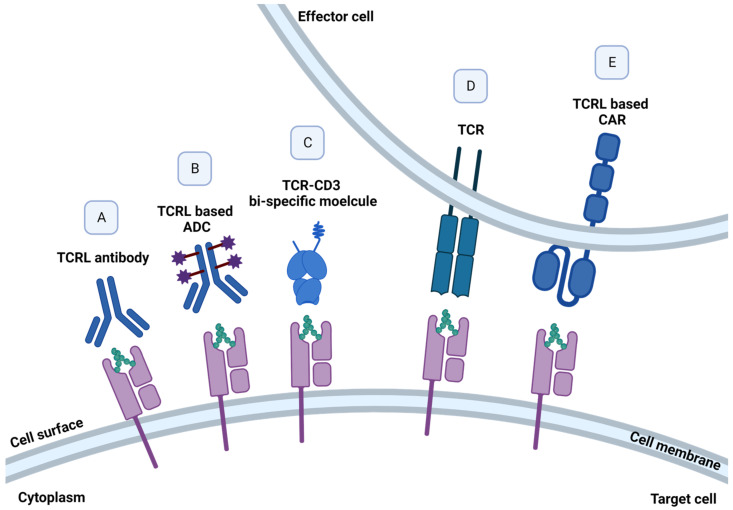
TCR- and TCRL-based structures. TCRL- and TCR-based cancer therapies, targeting pMHC molecules presented by malignant cells. Elimination of pMHC expressing cells can be achieved using (**A**) soluble naked full IgG TCRL antibody, mediating ADCC, CDC, or ADCP; (**B**) selective delivery of cytotoxic agents to tumor cells using soluble TCRL antibody–drug conjugates (ADCs); (**C**) fusion of TCRL and TCR to an anti-CD3 effector moiety, redirecting T cells against target malignant cells; and (**D**) engineered T cells, expressing specific TCR- and (**E**) TCRL-based CAR-T cells, programmed to recognize and induce cancer cell death.

**Table 1 cells-12-00027-t001:** Current TCR-based clinical trials.

Target	Cancer Type	Clinical Phase
MAGE-C2/HLA-A2	Melanoma and Head and Neck cancer	Phase I and II
HA-1	Relapsed or refectory and Acute Leukemia after donor stem cell transplant	Phase I
HBV	Related Hepatocellular Carcinoma in Post Liver Transplantation	Phase I
KRAS G12V	Pancreatic Cancer, Pancreatic Neoplasms, Pancreatic Ductal Adenocarcinoma and Advanced Cancer	Phase I and II
NY-ESO-1	Bone Sarcoma and Soft Tissue Sarcoma	Phase I
KK-LC-1	Gastric Cancer, Breast Cancer, Cervical Cancer and Lung Cancer	Phase I
Mesothelin	Metastatic Pancreatic Ductal Adenocarcinoma and Stage IV Pancreatic Cancer AJCC v8	Phase I
MCPyV-specific HLA-A02	Metastatic or Unresectable Merkel Cell Cancer	Phase I and II
MAGE-A1	Advanced Solid Tumors	Phase I and II
H3.3K27M	Diffuse Midline Glioma, H3 K27M-Mutant	Phase I
G12D variant of mutated Ras	Gastrointestinal Cancer, Pancreatic Cancer, Gastric Cancer, Colon Cancer and Rectal Cancer	Phase I and II
MAGE-A3/A6	Solid Tumor	Phase I

Source: https://clinicaltrials.gov (accessed on 15 October 2022).

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
