# Peer review of "The Intracellular Proteome as a Source for Novel Targets in CAR-T and T-Cell Engagers-Based Immunotherapy"

_cells, 2022, doi:10.3390/cells12010027_

Round 1

Reviewer 1 Report

In this manuscript, authors presented the selection of TCR and TCRL based targeting moieties, with pre-clinical and clinical studies, examining peptide-MHC targeting agents in cancer immunotherapy.

The Title of the manuscript is concise and relevant. The aim and scope of the study explained well. Introduction is quite comprehensive and highlighted work importance as well as its significance towards future prospective. Overall, this review is nicely written.

Before proceeding further, I expect the authors to thoroughly proofread the document and fix all grammatical and typographical errors (some examples include L8-9, L36, L40-41, L73, L78, L95, L101, L116 etc).

Minor suggestions:

1) Mention the reference source in Table 1.

2) Provide author’s contribution at the end.

Author Response

In this manuscript, authors presented the selection of TCR and TCRL based targeting moieties, with pre-clinical and clinical studies, examining peptide-MHC targeting agents in cancer immunotherapy.

The Title of the manuscript is concise and relevant. The aim and scope of the study explained well. Introduction is quite comprehensive and highlighted work importance as well as its significance towards future prospective. Overall, this review is nicely written.

The title of the manuscript was revised. Thanks for the comment.

Before proceeding further, I expect the authors to thoroughly proofread the document and fix all grammatical and typographical errors (some examples include L8-9, L36, L40-41, L73, L78, L95, L101, L116 etc).

Manuscript was fixed for grammatical and typographical errors including the indicated line. A comprehensive table with changes is attached.

Minor suggestions:

1) Mention the reference source in Table 1.

Reference was added

2) Provide author’s contribution at the end.

Authors contributions were added

Reviewer 2 Report

The review cover very interesting and timely topic. There are few linguistic and scientific mistakes that should be corrected.

The title is rather misleading, as the review also describes TCR-T cells and TCR-like antibodies in cancer immunotherapy.

Line 17: please define TCRL or use the dull name in the abstract

30: “non-self”: rather than “abnormal”

35: introducing “HLA heavy chain” is a bit misleading, as the HLA nomenclature only applies to humans,

80: depending on cell type, rather than “depends on”

81:“ immune cells experienced with antigen presentation” – professional antigen presenting cells

96: TCR on cell surface consists of alpha and beta chains/proteins, not genes

96: maybe “sequences” rather than clones

103: “in close proximity” sounds a bit vague, TCR forms complex with CD3 chains

140: “Alternatively, MHC class I molecules can  also be expressed using the two distinct heavy and β2M light chains, expressed by E. coli followed by refolding of the full tetrameric pMHC class I molecule with synthetic pep” This is incorrect. MHC class I refolding requires heavy chain, B2m and peptide, resulting in pMHC trimer.

158: it is not clear how human B cell cloning can be achieved if animals are vaccinated

193: is “predicate” used correctly here?

203: HLA-A2 a group of alleles, not a single allele; the one of the common alleles in populations of European ancestry is  A*02:01

Author Response

The review cover very interesting and timely topic. There are few linguistic and scientific mistakes that should be corrected.

The title is rather misleading, as the review also describes TCR-T cells and TCR-like antibodies in cancer immunotherapy.

The title was revised. Thanks for the comment.

Line 17: please define TCRL or use the dull name in the abstract

TCRL was defined in line 14 of the abstract

30: “non-self”: rather than “abnormal”

corrected

35: introducing “HLA heavy chain” is a bit misleading, as the HLA nomenclature only applies to humans,

corrected

80: depending on cell type, rather than “depends on”

Corrected

81:“ immune cells experienced with antigen presentation” – professional antigen presenting cells

corrected

96: TCR on cell surface consists of alpha and beta chains/proteins, not genes

revised

96: maybe “sequences” rather than clones

revised

103: “in close proximity” sounds a bit vague, TCR forms complex with CD3 chains

revised

140: “Alternatively, MHC class I molecules can  also be expressed using the two distinct heavy and β2M light chains, expressed by E. coli followed by refolding of the full tetrameric pMHC class I molecule with synthetic pep” This is incorrect. MHC class I refolding requires heavy chain, B2m and peptide, resulting in pMHC trimer.

Revised

158: it is not clear how human B cell cloning can be achieved if animals are vaccinated

Revised

193: is “predicate” used correctly here?

corrected

203: HLA-A2 a group of alleles, not a single allele; the one of the common alleles in populations of European ancestry is  A*02:01

revised

Attached is a table with all change made in the manuscript
